# A Robust and Comprehensive Study of the Molecular and Genetic Basis of Neurodevelopmental Delay in a Sample of 3244 Patients, Evaluated by Exome Analysis in a Latin Population

**DOI:** 10.3390/diagnostics15030376

**Published:** 2025-02-05

**Authors:** Julian Lamilla, Taryn A. Castro-Cuesta, Paula Rueda-Gaitán, Laura Camila Rios Pinto, Diego Alejandro Rodríguez Gutiérrez, Yuri Natalia Sanchez Rubio, Carlos Estrada-Serrato, Olga Londoño, Cynthia Rucinski, Mauricio Arcos-Burgos, Mario Isaza-Ruget, Juan Javier López Rivera

**Affiliations:** 1Laboratorio Clínico Especializado, Clínica Universitaria Colombia, Clínica Colsanitas, Bogotá 111321, Colombia; castrotaryn@gmail.com (T.A.C.-C.); pauarueda@colsanitas.com (P.R.-G.); lacrios@colsanitas.com (L.C.R.P.); diegoalerodriguez@colsanitas.com (D.A.R.G.); yuri.sanchez@colsanitas.com (Y.N.S.R.); carestrada@keralty.co (C.E.-S.); olgalondono0821@gmail.com (O.L.); crucinskic@unal.edu.co (C.R.); mauricio.arcos@udea.edu.co (M.A.-B.); 2Grupo de Genética Médica, Clínica Universitaria Colombia, Clínica Colsanitas, Bogotá 111321, Colombia; misaza@keralty.com; 3Instituto de Investigaciones Médicas, Facultad de Medicina, Universidad de Antioquia, Medellín 050010, Colombia; 4Keralty, Sanitas International Organization, Grupo de Investigación INPAC, Fundación Universitaria Sanitas, Bogotá 110131, Colombia

**Keywords:** neurodevelopmental disorders, whole exome sequencing, genetic testing

## Abstract

**Background and Objectives**: Neurodevelopmental disorders (NDDs), including developmental delay (DD), autism spectrum disorder (ASD), intellectual disability (ID), attention-deficit/hyperactivity disorder (ADHD), and specific learning disorders, affect 15% of children and adolescents worldwide. Advances in next-generation sequencing, particularly whole exome sequencing (WES), have improved the understanding of NDD genetics. **Methodology**: This study analyzed 3244 patients undergoing WES (single, duo, trio analyses), with 1028 meeting inclusion criteria (67% male; aged 0–50 years). **Results**: Pathogenic (P) or likely pathogenic (LP) variants were identified in 190 patients, achieving a diagnostic yield of 13.4% (singleton), 14% (duo), and 21.2% (trio). A total of 207 P/LP variants were identified in NDD-associated genes: 38% were missense (48 de novo), 29% frameshift (26 de novo), 21% nonsense (14 de novo), 11% splicing site (14 de novo), and 1% inframe (1 de novo). De novo variants accounted for 49.8% of cases, with 86 novels de novo variants and 27 novel non de novo variants unreported in databases like ClinVar or scientific literature. **Conclusions**: This is the largest study on WES in Colombian children with NDDs and one of the largest in Latino populations. It highlights WES as a cost-effective first-tier diagnostic tool in low-income settings, reducing diagnostic timelines and improving clinical care. These findings underscore the feasibility of implementing WES in underserved populations and contribute significantly to understanding NDD genetics, identifying novel variants with potential for further research and clinical applications.

## 1. Introduction

Neurodevelopmental disorders (NDDs) are a heterogeneous group of conditions that affect brain development. These disorders, including developmental delay (DD), autism spectrum disorder (ASD), intellectual disability (ID), attention-deficit/hyperactivity disorder (ADHD), and specific learning disorders, affect approximately 15% of children and adolescents worldwide [1,2]. The etiology of NDDs is complex, involving both genetic and environmental factors. Recent advances in genetic research, particularly next-generation sequencing technologies, have significantly enhanced our understanding of the genetic landscape underlying these disorders, enlightening us about the intricate mechanisms at play [2,3,4].

The genetic architecture of NDDs exhibits extensive locus heterogeneity and a spectrum of variant types, including single-nucleotide variants (SNVs), copy number variations (CNVs), and structural variations. Large-scale exome sequencing studies have revealed that de novo and inherited rare variants contribute substantially to individual risk for NDDs [4,5]. For instance, studies have identified over 100 high-confidence ASD-associated genes enriched with likely deleterious de novo variants. However, the genetic landscape is incomplete, with estimates suggesting that up to 1000 genes may harbour de novo variants in ASD alone [5].

Recent meta-analyses and large-scale studies have provided important insights into the heritability and genetic overlap of various NDDs [2]. Family-based studies indicate that approximately two-thirds of the variation in NDDs can be attributed to genetic differences between individuals [3]. Moreover, there is significant genetic overlap between ASD, ID, ADHD, and other neurodevelopmental conditions such as epilepsy [2,3]. For example, substantial genetic correlations have been observed between ASD and ADHD and between communication disorders and specific learning disorders [3,6].

As mentioned before, whole exome sequencing (WES) has emerged as a powerful first-tier diagnostic tool. In a previous study of our group, we reported 30–43% diagnostic yields for unexplained NDDs, representing a significant improvement over traditional diagnostic methods such as chromosomal microarray analysis [7]. Molecular diagnosis follow-up of the clinical progression and associated phenotypes could have profound implications for clinical management, including initiating targeted surveillance and providing accurate genetic counselling for families [2,7]. As our understanding of the genetic basis of NDDs continues to evolve, it is essential to identify overlapping phenotypes because it could help develop a more personalized therapeutic approach in the future, offering hope for more effective treatments [1,5,7].

## 2. Materials and Methods

### 2.1. Patient Recruitment and Sampling

Patients attended consultation for medical genetics at Clínica Colsanitas between January 2021 and October 2023. The inclusion criteria were the following: (1) patients with suspected or non-genetic clinical diagnosis of neurodevelopmental disorders (NDDs), recorded in terms of HPO (Human Phenotype Oncology) or the Diagnostic and Statistical Manual of Mental Disorders, DSM-5; (2) patients with requests for whole exome sequencing (WES) in trio (WES patient and both parents; 141 patients), duo (WES patient and one of his parents; 9 patients) or singleton (WES patient; 40 patients). The following patients were excluded from the study: (1) patients with identification of NDDs caused by an extrinsic event such as perinatal noxa, dystocic delivery, severe childhood or adolescent trauma with clear evidence of structural injury, complicated infection, and neurological symptomatic such as localized or generalized meningoencephalitis, a clear history of perinatal or childhood and adolescent anoxia or hypoxia due to a traumatic event; (2) patients with clear and defined metabolopathy of the perinatal stage with severe neonatal or perinatal distress leading to neurological deterioration; (3) patients with known syndromes, defined by aneuploidies identified in the cytogenetic analysis.

### 2.2. Review of Medical Records

The research group extracted patients’ information from electronic medical records, and results were recorded in the information system of the Specialized Laboratory of Clínica Colsanitas. The information collected consisted of perinatal history, detailed developmental milestones, clinical suspicion or diagnosis of the patient, sex, age, personal history, imaging and electrophysiological test results, family history, clinical laboratory results, and genetic tests. Based on the clinical histories, the patients were categorized into the following phenotypes associated with NDD: neurodevelopmental delay (DD), autism spectrum disorder (ASD), cognitive impairment (CI), and neurodevelopmental delay + epilepsy (DD+Epilepsy).

### 2.3. Sampling, Sequencing, Bioinformatics Processing, and Variant Filtering

WES (clinical exome or trio-exome sequencing) was performed in the Specialized Laboratory of Clinica Colsanitas from DNA extracted from peripheral blood by next-generation sequencing (NGS), using Capture Probes targeting exomic regions based on Illumina DNA prep with enrichment^®^ and MGIEasy Exome Capture V5 Probe Set^®^. Sequencing was performed on NextSeq 2000 (Illumina, San Diego, CA, USA), NovaSeq 6000 Sequencing System (Illumina, San Diego, CA, USA) or G-400 (MGI Tech Co., San Jose, CA, USA). Sequence reads were aligned with ConsortiumHuman Build 37, and visualization and variant identification were performed with the SOPHiA DDM and VarSome Clinical platforms. This methodology allows detection of single-nucleotide variants (SNVs) and insertions/deletions (Indel). The genetic variants found were classified as pathogenic (P), likely pathogenic (LP), variant of uncertain significance (VUS), likely benign (LB) or benign (B) according to the guidelines of the American College of Medical Genetics and Genomics (ACMG), supported by different clinical databases such as ClinVar from NCBI (https://www.ncbi.nlm.nih.gov/clinvar/ (accessed on 12 June 2024)), ClinGen (https://clinicalgenome.org/ (accessed on 12 June 2024)), GnomAD (https://gnomad.broadinstitute.org/ (accessed on 12 June 2024)), Franklin (https://franklin.genoox.com/clinical-db/home (accessed on 13 June 2024)), UniProt (https://uniprot.org/ (accessed on 13 June 2024)), and OMIM (https://omim.org/ (accessed on 12 June 2024)). Variant analyses in inpatients with NDDs were performed as follows: single or panel (patient only analyzed), duo (patient and one parent analyzed), and trio (patient and both parents analyzed).

### 2.4. Classification and Ontology Analysis of Associated Genes

The identified genes associated with NDDs were classified into four groups based on patient history and clinical diagnosis. Genes and patients without specific phenotypes were classified into neurodevelopmental delay. To demonstrate the biological significance of the genes associated with the four (ASD, CI, DD, and DD and epilepsy) phenotypic groups, gene ontology network analysis was performed using Cytoscape software (v.3.10.2) with the ClueGo plug-in (v.2.5.10). With ClueGo, enrichment of the studied genes with GO terms linked to the kappa score is performed. Only GO terms with *p*-values < 0.05 were adjusted to the Bonferroni step-down.

### 2.5. Statistical Analysis

Qualitative variables are presented as absolute and relative frequencies. Quantitative variables are described with measures of central tendency and dispersion. To calculate the diagnostic yield, a positive result was considered to be any result that had a variant classified as pathogenic (P) or likely pathogenic (LP) in a gene associated with a phenotype that correlates clinically with the phenotype of the patient under study (gene–disease association was performed according to the OMIM database). All results in which no variants related to the patient’s phenotype were reported were considered harmful. The diagnostic yield of the test was calculated considering the total number of patients included in the study, which corresponds to the percentage of patients with a positive result.

### 2.6. Ethical Considerations

The present study was developed within the framework of the macro research entitled “Development of a comprehensive care model based on personalised medicine for the diagnosis, treatment and follow-up of pediatric patients with NDD in the Colombian population”, which was reviewed and approved by the Research Ethics Committee of the Fundación Universitaria Sanitas (CEIFUS code 1419-21, date of approval, 2 July 2021), following the principles of the Declaration of Helsinki. Clinica Colsanitas have approved the informed consent form for this study, and the participants, or their parents/legal representatives, have signed the corresponding informed consent form to perform WES.

The information associated with the patients above was handled exclusively by the principal investigators, who always respected the provisions of law 1581 of Colombian legislation regarding the handling of personal data.

## 3. Results

Between January 2021 and October 2023, a total of 3244 patients with single, duo, and trio exome analyses were sequenced by NGS. In total, 1028 (31.7%) met the inclusion criteria; 67% (688) were male, and 33% (340) were female. The age range of patients associated with NDD was between 0 and 50 years of age (mean $\overset{-}{X}$: 5 SD: 4.7). Of the 1028 patients who met the NDD criteria, 695 patients had clinically suspected DD (68% of patients), 199 had ASD (19% of patients), 79 had DD and epilepsy (8% of patients), and finally, 55 (5% of patients) had CI.

### 3.1. Diagnostic Yield

Pathogenic (P) or likely pathogenic (LP) variants were identified in 190 patients, corresponding to 18.5% of patients with NDD admissions and 5.6% of the total patients analyzed in this period. The diagnostic yield varied depending on the type of analysis performed: clinical exome 13.4% (299 admissions/40 positive), duo WES 14% (63 admissions/9 positive), and trio WES 21.2% (666 admissions/141 positive) (Figure 1).

A total of 139 genes associated with NDDs were identified in 190 patients with P/LP variants, of which 116 genes were identified in 151 patients with DD, 16 genes in 22 patients with DD and epilepsy, 13 genes in 12 patients with ASD, and 7 genes in 5 patients with CI (Table 1). Some genes were associated with more than one of the above phenotypes (Figure 2). In total, 186 P/LP variants were identified in the 190 NDD-positive patients (Appendix A). The majority of patients (78%, 146/186) presented SNV-type variants associated with genes with autosomal dominant (AD) inheritance patterns, followed by 9% (17/186) of patients with genes with autosomal recessive (AR) inheritance patterns. Finally, genes with X-linked dominant (XLD), X-linked recessive (XLR), and X-linked (XL) inheritance patterns were identified in 8% (14/186), 3% (6/186), and 2% (3/186) of patients, respectively. In total, 82% (156/190) of patients had variants in a heterozygous state, 8% (15/190) homozygous, 6% (11/190) compound heterozygous, and 4% (8/190) hemizygous.

### 3.2. Characterization of Novel Variants

A total of 207 P/LP SNVs were identified in NDD-associated genes in 190 patients, and missense variants were the most represented with 78 findings (38%; 48 de novo); 61 (29%) frameshift (26 de novo); 44 (21%) nonsense (14 de novo); 22 (11%) splicing site (14 de novo); two (1%) inframe (one de novo) (the list of all variants can be found in the Appendix A). Finally, 49.8% (103/207) of the variants were identified de novo in 100 patients with NDDs. Additionally, we report 86 new variants de novo that have not been listed in ClinVar nor reported previously in any of the databases consulted or in scientific articles (marked with * in Table 2). In addition, 27 non de novo SNVs had not been previously reported in scientific articles or databases (marked without * in Table 2). However, it was possible to classify them as P or LP according to ACMG and in silico analysis using relevant databases and tools (Table 2).

Genes with the highest number of variants were *MECP2* (n = 6) and *CUL3* (n = 4). Two nonsense and two frameshifts were identified in this gene; three were de novo (c.764C>G, p.Ser255*; c.769delG, p.Glu257Lysfs*5; c.494dup, p.Leu166Ilefs*37), and two were not previously reported (769delG, p.Glu257Lysfs*5; c.494dup, p.Leu166Ilefs*37).

### 3.3. Gene Ontology Analysis

Through GO network analysis of biological processes, based on the four categories in which the NDD genes were classified, the genes associated with ASD, CI, and DD presented different patterns among them. In the case of the genes related to DD and epilepsy, they did not show any pattern of association. Three ASD genes and four CI genes presented significant association (false discovery rate and *p*-value < 0.05). IC genes showed a single association with the term GO protein demethylation and ASD genes with social behaviour (Figure 3A,B). As for the DD genes, 113 genes formed a complex network without a solid or consistent association with each other. Some fragmented associations were observed, such as protein methylation (12 genes), telencephalon development (12 genes), associative learning (9 genes), regulation of cell cycle G1/S phase transition (8 genes), positive regulation of type I interferon production (6 genes), glial cell migration (5 genes), face development (5 genes), negative regulation of cell–matrix adhesion (5 genes), negative regulation of cell–matrix adhesion (5 genes), negative regulation of cell–substrate adhesion (5 genes), DNA duplex unwinding (5 genes), axo-dendritic transport (5 genes), head morphogenesis (4 genes), voltage-gated sodium channel activity (4 genes), and protein destabilization (4 genes) (Figure 3C).

## 4. Discussion

Despite the genomic analyses, understanding the etiology of NDDs remains challenging due to their broad genetic and phenotypic heterogeneity. WES has played an essential role in identifying causatives and has proven to be a valuable diagnostic tool. This study achieved a molecular diagnosis for 190/1028 patients with NDD spectrum in Clínica Colsanitas between January 2021 and October 2023.

In patients with NDD and the complete spectrum of ASD, ID, and DD, we applied a retrospective chart review of the probands’ and their relatives’ medical records before continuing to the molecular analyses. The epilepsy phenotype was excluded. It was taken into account for the results but omitted from the analyses as it is an extensive phenotype and will be considered for future group publications.

Clinical studies should address the effectiveness of genetic studies targeting the aetiological diagnosis. The NDD spectrum is highly heritable and heterogeneous and affects a significant proportion of the population. The causes of these disorders may be linked to environmental factors like malnutrition, infections during pregnancy, drug misuse, or pollution through epigenetic dysregulation. Additionally, synaptopathies are also a significant cause of NDDs affecting brain plasticity, signalling, and connectivity, leading to an imbalance between excitatory and inhibitory signals in the brain. It is also known that differentially expressed gene networks enriched neurotransmitter and synapse activity, immune processes, and cortical development [6]. Achieving a molecular diagnosis of NDD has an economic impact not only on the patient’s healthcare system but also on their families.

The array comparative genomic hybridization (aCGH) has supported greater diagnostic effectiveness concerning historical cytogenetic techniques (3 vs. 10%, respectively) [8]. Development of next-generation sequencing (NGS) techniques, which allow genome sequencing (GS), or WES, have shown a high diagnostic capacity, leading to an exponential drop in costs and expanded sequencing coverage of the genome, as well as the ability to capture high read depth to detect low-level mutations. In our study, the diagnostic yield of trio-exome sequencing was significantly high (21.2%) and like reported yields in other studies [7,8], supporting its utility in molecular diagnosis of NDD spectrum patients. The strengths of this study are the sizable Colombian cohort, which included all patients assessed by a clinical geneticist, and the extensive clinical and phenotypic data that were available, which were used for the stratification of the central overlapping NDD spectrum phenotype.

The most common clinical symptom shared by all NDDs was cognitive dysfunction and autism. The classification of patients in the diagnostic spectrum categories might have also affected the proportion of ASD; patients with ASD features were diagnosed as DD due to either not being old enough to be diagnosed as ASD or not being assessed with a specific ASD-standardized scale to provide an accurate diagnosis. There was statistical significance between molecular findings on each phenotype, ID/NDD, ASD/NDD, and DD/NDD.

Evidence of diagnostic yield supports NGS-based genetic testing for diagnosing NDDs [2,8,9,10]. Current evidence suggests that a diagnostic yield of 35% can be obtained for WES in patients with intellectual disability and global developmental delay and 15% for patients with ASD [11]. The overall WES yield for NDD was 36%, 31% for isolated DD, and 53% for NDD with associated conditions. In the present study, an overall diagnostic rate of 18.5% was observed for WES, within the range reported in the literature [12]. Additionally, when evaluating only patients with clinical suspicion of DD, the diagnostic yield was 22% (151/695), ASD was 6% (12/199), DD and epilepsy was 28% (22/79), and IC was 5% (5/55).

The diagnostic yield was significantly higher for trio exome analysis (21.2%) compared to single clinical exome (13.4%). This supports using whole exome sequencing (WES) as an initial diagnostic tool for neurodevelopmental disorders (NDDs), surpassing targeted molecular studies like FMR1 triplet expansion or aCGH [12,13,14].

Mutations in various genes have been shown to interfere with the proliferation and migration of neurons. Genetic advances resulting from new sequencing techniques have also expanded our understanding of neuronal migration disorders that affect each stage of neurodevelopment. From the early stages of gestation, the development of the neuroepithelial progenitors that cover the wall of the ventricles begins. This process requires different stages, such as proliferation, differentiation, and migration. Enhancing corticogenesis is characterized by several unique features, including a unique germinal zone and the outer subventricular zone, increasing in size from week 20, and making an integration circuit [15,16]. Pathogenic variants that could modify cortical development and the timing and complexity of cortical neurogenesis or synaptogenesis could be linked to neurodevelopmental disorders, providing evidence for their physiological relevance [17,18].

We report the mutational spectrum of *MECP2*, *CUL3*, and *SHANK3* genes. The *MECP2* (MIM*300005) gene binds methylated CpGs, is a chromatin-associated protein required for mature neurons and is developmentally regulated. The MECP2 syndrome is a neurodevelopmental disorder that occurs almost exclusively in females and has an XLD inheritance pattern. It is characterized by arrested development between 6 and 18 months of age, regression of acquired skills, loss of speech, stereotypic movements (classically of the hands), microcephaly, seizures, and cognitive impairment. Also, it is associated with susceptibility to X-linked autism 3, with an XL inheritance pattern. Pathogenic and probably pathogenic variants in homozygous or compound heterozygous state in the *MECP2* gene have also been associated with severe neonatal encephalopathy, syndromic X-linked intellectual development disorder 13 and syndromic X-linked intellectual development disorder, Lubs type, phenotypes with XLR inheritance pattern [19]. Two nonsense and three missense variants were identified (one present in two patients).

The *CUL3* gene (MIM*603136) encodes a scaffolding component of the Cullin-RING ligase (CRL) complex, essential for mitotic division. CUL3 transcript levels are relatively high during early embryonic development, playing an important role during fetal development and maturation. CUL3 neurodevelopmental disorder with or without autism or seizures, with autosomal dominant inheritance pattern, is characterized by global developmental delay evident in childhood, impaired intellectual development, and speech delay. Some patients develop seizures and may show regression after the onset of seizures. Others present with autistic features or behavioural abnormalities. Additional variable systemic features such as cardiac defects, growth retardation or brain imaging abnormalities may also be present. Also, it is associated with pseudohypoaldosteronism type IIE, which is related to an AD inheritance pattern [20,21].

The *SHANK3* gene (MIM*606230), expressed predominantly in the cerebral cortex and cerebellum, encodes a scaffolding protein that is enriched in postsynaptic densities of excitatory synapses; it has been shown to bind to neuroligins, which, together with the neurexins, form a complex at glutamatergic synapses. *SHANK3* was shown to coincide with the most severe cases of autism and Phelan–McDermid syndrome (22q13.3 deletion syndrome when including the gene), because it affects the development and morphology of dendritic spines and reduces synaptic transmission in mature neurons, contributing to an imbalance of inhibition to excitation [22].

Because *CUL3* mutations disrupt protein degradation pathways, accumulating misfolded or damaged proteins and impairing neuronal health, and SHANK mutations impair synaptic signalling and structural integrity, directly impacting neural communication, supplementing with antioxidants like N-acetylcysteine (NAC) to counteract cellular damage linked to CUL3 dysfunction, as well as the use of pharmacological agents like mGluR modulators (e.g., mGluR5 antagonists) to balance excitatory/inhibitory signalling disrupted by the *SHANK* mutations, might be of potential use in personalized therapeutic scenarios [21].

In positive trio exomes, 100 LP/P variants were identified as de novo (absent in the parents). De novo variants gain value and increase the diagnostic performance of the test when there is no family history of similar conditions or reports of consanguinity [20]. However, it is essential to highlight that in cases where variants inherited from the parents are identified, the inheritance pattern of the phenotype should be considered since there are cases where the variants can be heterozygous in the mother, being carriers, and hemizygous in male children, who would express the phenotype for diseases with recessive patterns linked to the X chromosome. Another possibility is that there are diseases for which incomplete penetrance and variable expressivity have been reported due to differences in genetic background, environmental factors, or a combination of both, so it is possible that within the same family, there are affected individuals with different levels of involvement, or even that they are asymptomatic despite having the same variant [23,24].

The present study has several limitations, such as its retrospective design, the possible overestimation of the clinical exome sequencing diagnostic yield due to a bias in the selection of samples for NGS, and the fact that aCGH is not being compared with WES. To confirm our results, a prospective study comparing the diagnostic yield of aCGH and clinical exome sequencing in an unbiased sample would be desirable, and for those recently described new variants, further mechanistic and phenotypic characterization of additional patients could confirm their roles in human neurodevelopment disease and to delineate their associated phenotypic spectrums.

To our knowledge, this is the first study to evaluate the clinical utility of WES for children with NDD spectrum in Colombia in a large cohort of patients; we show the importance of reducing the expenses in genetic testing for NDDs with a cost-saving first-tier diagnostic test that would serve for developing countries, and to define an aetiological diagnosis in less time.

## Figures and Tables

**Figure 1 diagnostics-15-00376-f001:**
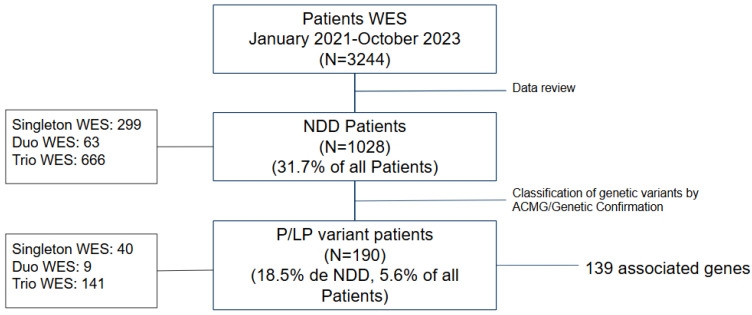
Summary of the sample of patients according to (1) the presence of pathogenic (P) of likely pathogenic (LP) variants in agreement with the ACMG criteria, and (2) if the exome sequencing was applied to a singleton, duo, or trio unit of analysis.

**Figure 2 diagnostics-15-00376-f002:**
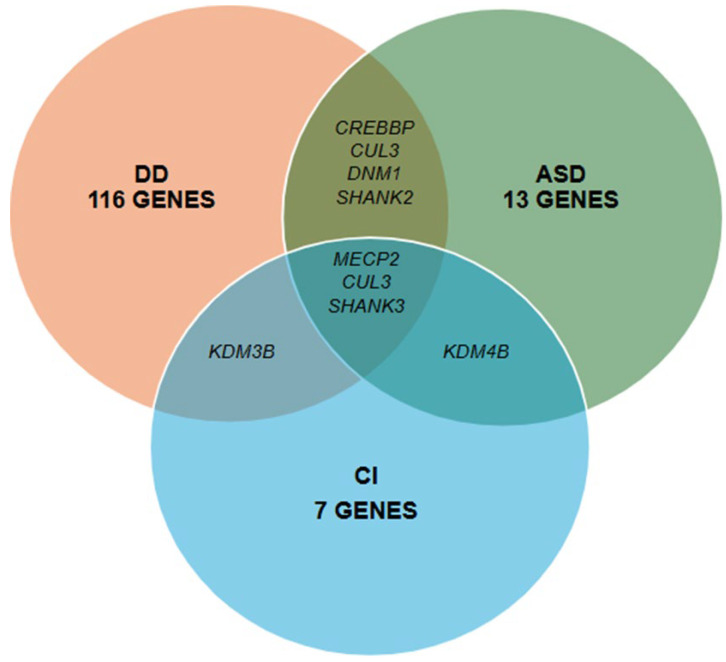
Genes with LP/P variants overlapped between phenotypes. DD, ASD, and CI variants in the *MECP2*, *CUL3*, and *SHANK3* genes were described. ASD and CI variants were described on DD and CI in KDM4B and KDM3B variants.

**Figure 3 diagnostics-15-00376-f003:**
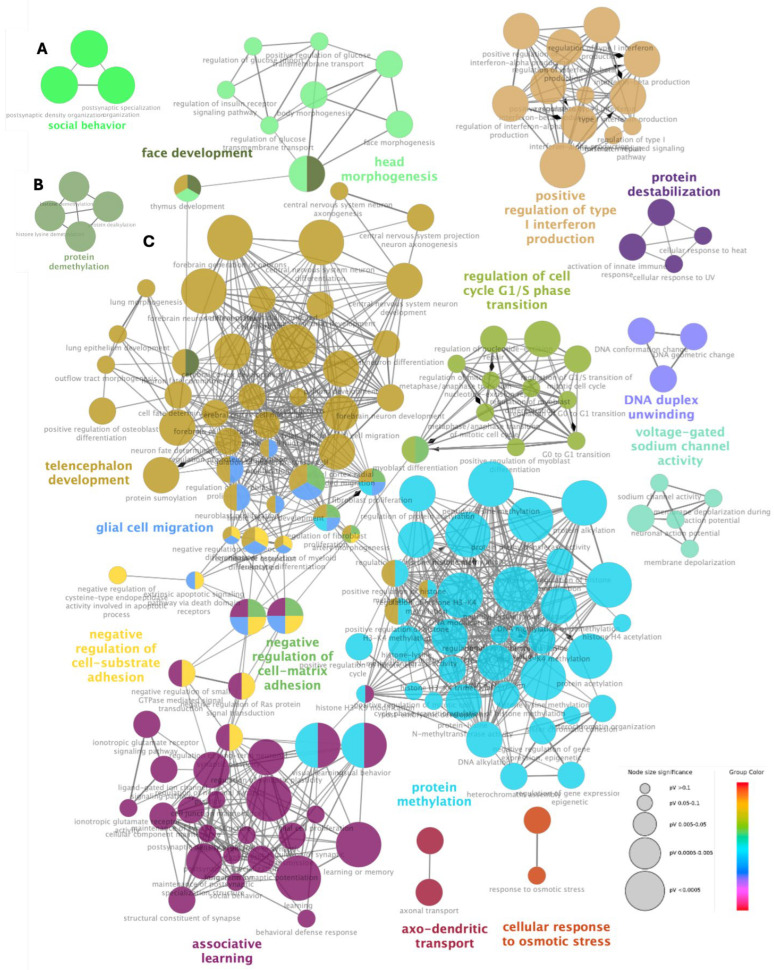
Functionally clustered gene networks for the phenotype groups. (**A**) Autism spectrum disorder (ASD) with a single association with social behaviour. (**B**) Cognitive impairment (CI) with a single association with protein demethylation. (**C**) Neurodevelopmental delay (DD) forms a complex network, without solid association composed of 15 fragmented associations. The networks were obtained from ClueGo enrichment analysis. Gene ontology terms and associated genes are shown in the same colour. The node size of each term corresponds to its importance in the enrichment.

**Table 1 diagnostics-15-00376-t001:** Genes in which LP/P variants were identified by phenotype.

Phenotype	Gene
Developmental delay	*ACTL6B*, *AHDC1*, *ANKRD11*, *ARID2*, *ATRX*, *BCL11A*, *BCL11B*, *BRAF*, *CHD3*, *CREBBP*, *CTNNB1*, *CUL3*, *DDX3X*, *DNM1*, *DNMT3A*, *DPYD*, *DYNC1H1*, *FBXO11*, *FLNB*, *FOXG1*, *GLUD2*, *GNB1*, *GRIA2*, *GRIN1*, *H1-4*, *IFIH1*, *INTS1*, *IVD*, *JAG1*, *KCNT2*, *KDM6B*, *KIAA1109*, *KIF1A*, *KMT2A*, *KMT2C*, *LARP7*, *LARS2*, *LMAN2L*, *LZTR1*, *MECP2*, *MN1*, *MSTO1*, *NACC1*, *NALCN*, *NARS*, *NEXMIF*, *NF1*, *NFIB*, *PHF8*, *PIK3R1*, *PPM1D*, *PTPN11*, *PURA*, *QRICH1*, *RAF1*, *RAI1*, *RBMX*, *RPS6KA3*, *SATB2*, *SCN1A*, *SCN2A*, *SCN8A*, *SETD2*, *SETD5*, *SHANK2*, *SHANK3*, *SIX3*, *SLC13A5*, *SPEN*, *STAG2*, *SYNGAP1*, *TAOK1*, *TBK1*, *TBR1*, *TCF20*, *TCF4*, *TOE1*, *TREX1*, *TRIP12*, *TSC2*, *TSPAN7*, *TUBB*, *UBTF*, *USP7*, *WAC*, *ZMIZ1*, *AP1S2*, *ARSA*, *CAPN3*, *CDC42*, *COL6A2*, *COL6A3*, *CTBP1*, *DCX*, *DHTKD1*, *EP300*, *FBN2*, *GCDH*, *PMM2*, *POLR3B*, *PUF60*, *RTN2*, *RYR1*, *SPG11*, *SPG7*, *SPTBN2*, *TNPO3*, *TRPV4*, *TRIO*, *RNASEH2B*, *RFX7*, *KMT2E*, *ATP7B*, *ARID1B*, *INPP5E*, *KDM3B*, *CSNK2A1*
Developmental delay and epilepsy	*ABCC8*, *AFF3*, *ANKRD17*, *CDKL5*, *CHD2*, *GNAO1*, *GRIN2B*, *HNRNPU*, *KAT6A*, *KCNB1*, *KCNMA1*, *PTCH1*, *SETD1A*, *SLC2A1*, *STXBP1*, *MN1*
Autism spectrum disorder	*CACNA1E*, *CAMK2A*, *CHD8*, *CPT2*, *CREBBP*, *CSF1R*, *CUL3*, *DNM1*, *EBF3*, *KDM4B*, *PTEN*, *SHANK2*, *SHANK3*
Cognitive impairment	*KDM4B*, *KDM5C*, *KDM6A*, *PAX8*, *NAA15*, *KDM3B*, *SHANK3*

**Table 2 diagnostics-15-00376-t002:** Description of variants without previous reports in the literature or databases. **Note:** Those variants marked with * were found de novo, and variants marked with # were identified in the patient and one of the parents.

Gene	DNA	Protein	Variant Type	ACMG Classification	Transcripts	OMIM Code
*ACTL6B* ^#^	c.521_522insA	p.Thr175Hisfs*7	Frameshift	LP (PVS1, PM2)	NM_016188.4	612458
*ACTL6B* *	c.991G>A	p.Gly331Ser	Missense	LP (PS2, PM2, PP2)	NM_016188.5	612458
*AHDC1*	c.4294del	p.Ala1432Profs*13	Frameshift	LP (PVS1, PM2)	NM_001371928.1	615790
*ANKRD11*	c.741C>A	p.Tyr247*	Nonsense	LP (PVS1, PM2)	NM_013275.6	611192
*ANKRD11* *	c.7124_7152del	p.Glu2375Alafs*147	Frameshift	P (PVS1, PS2, PM2)	NM_013275.5	611192
*ANKRD17* *	c.4453_4457del	p.Lys1485Glufs*17	Frameshift	P (PVS1, PS2, PM2)	NM_032217.4	615929
*AP1S2* ^#^	c.180-1G>C		Splicing site	LP (PVS1, PM2)	NM_001272071.2	300629
*BCL11B* *	c.2345dup	p.Gly785Argfs*100	Frameshift	P (PVS1, PS2, PM2)	NM_138576.3	606558
*BRAF* *	c.2030A>G	p.Asp677Gly	Missense	P (PS2, PM1, PM2, PP2, PP3)	NM_004333.6	164757
*CACNA1E* *	c.5365-2A>G	-	Splicing site	LP (PS2, PM2)	NM_001205293.1	601013
*CHD2* *	c.2822A>T	p.Gln941Leu	Missense	LP (PM2, PM6, PP2, PP3)	NM_001271.4	602119
*CHD3*	c.3481C>T	p.His1161Tyr	Missense	LP (PM1, PM2, PP2, PP3)	NM_001005273.3	7806365
*CHD8* *	c.4987_5003del	p.Val1663Glnfs*59	Frameshift	P (PVS1, PS2, PM2)	NM_020920.4	610528
*COL6A3* *	c.6156+2T>C	-	Splicing site	P (PVS1, PS2, PM2, PP3)	NM_004369.4	120250
*COL9A3* ^#^	c.1021C>T	p.Arg341*	Nonsense	LP (PVS1, PM2)	NM_001853.4	120270
*CUL3* *	c.769delG	p.Glu257Lysfs*5	Frameshift	P (PVS1, PS2, PM2)	NM_003590	603136
*CUL3* *	c.494dup	p.Leu166Ilefs*37	Frameshift	P (PVS1, PS2, PM2)	NM_003590.4	603136
*DCX*	c.166C>G	p.Arg56Gly	Missense	LP (PM1, PM2, PM5, PP2, PP3)	NM_001195553.1	300121
*DDX3X*	c.1646A>T	p.Asn549Ile	Missense	LP (PM1, PM2, PP2, PP3)	NM_001356.4	300160
*DNM1* *	c.1751A>T	p.His584Leu	Missense	LP (PS2, PM2)	NM_004408.4	602377
*DNM1* *	c.2318+2T>C	-	Splicing site	P (PVS1, PS2, PM2)	NM_004408.4	602377
*DYNC1H1* *	c.11632C>G	p.Gln3878Glu	Missense	LP (PS2, PM2)	NM_001376.5	600112
*EP300* *	c.4779+1G>A	-	Splicing site	P (PVS1, PS2, PM2)	NM_001429.3	602700
*FBXO11* *	c.1685A>G	p.Tyr562Cys	Missense	LP (PS2, PM1, PM2, PP2, PP3)	NM_025133.4	607871
*GNB1* *	c.310G>C	p.Ala104Pro	Missense	LP (PS2, PM2, PP3)	NM_002074.5	139380
*GRIN1* *	c.2248G>A	p.Gly750Arg	Missense	LP (PS2, PM2, PP2, PP3)	NM_000832.7	138249
*GRIN1* *	c.1824G>C	p.Trp608Cys	Missense	P (PS2, PM1, PM2, PP2, PP3)	NM_007327.3	138249
*GRIN2B* *	c.1990T>C	p.Ser664Pro	Missense	LP (PS2, PM2, PP3)	NM_000834.3	138252
*HNRNPU*	c.1484_1487del	p.Lys495Ilefs*5	Frameshift	LP (PVS1, PM2)	NM_031844.3	602869
*HNRNPU* *	c.1576_1580del	p.Asn526Serfs*9	Frameshift	P (PVS1, PS2, PM2)	NM_031844.2	602869
*IFIH1* ^#^	c.2863C>T	p.Gln955*	Nonsense	LP (PVS1, PM2)	NM_022168.3	606951
*JAG1*	c.1725_1726dupTG	p.Asp576Valfs*168	Frameshift	LP (PVS1, PM2)	NM_000214.3	601920
*KAT6A* *	c.1019del	p.Asn340Thrfs*3	Frameshift	P (PVS1, PS2, PM2)	NM_006766.4	601408
*KAT6A* *	c.4140_4141insA	p.Asp1381Argfs*13	Frameshift	P (PVS1, PS2, PM2)	NM_006766.4	601408
*KCNB1* *	c.1223C>T	p.Pro408Leu	Missense	LP (PS2, PM1, PM2, PP3)	NM_004975.3	600397
*KCNB1* *	c.1202G>T	p.Gly401Val	Missense	P (PS2, PM1, PM2, PM5, PP3)	NM_004975.4	600397
*KCNMA1* ^#^	c.2095A>T	p.Lys699*	Nonsense	LP (PVS1, PM2)	NM_001161352.1	600150
*KCNT2*	c.3118C>T	p.Arg1040*	Nonsense	LP (PVS1, PM2)	NM_198503	610044
*KDM3B* *	c.3638dup	p.(Asp1214Ter	Nonsense	LP (PS2 PM2)	NM_016604.4	609373
*KDM4B*	c.2147del	p.Leu716Tyrfs*42	Frameshift	LP (PVS1, PM2)	NM_015015.2	609765
*KDM5C* *	c.1571A>T	p.Asn524Ile	Missense	LP (PM1, PM2, PM6, PP3)	NM_004187.5	314690
*KIAA1109* ^#^	c.4118_4119del	p.Ser1373*	Nonsense	LP (PVS1, PM2)	NM_015312.3	611565
*KIAA1109* *	c.18T>A	p.Asn6Lys	Missense	LP (PS2, PM2)	NM_015312.3	611565
*KIAA1109* *	c.4118_4119del	p.Ser1373*	Nonsense	LP (PVS1, PM2)	NM_015312.3	611565
*KMT2A* *	c.7187_7188del	p.Pro2396Argfs*2	Frameshift	P (PVS1, PS2, PM2)	NM_001197104.2	159555
*KMT2C* *	c.5551C>T	p.Gln1851*	Nonsense	P (PVS1, PS2, PM2)	NM_170606.2	606833
*KMT2E* ^#^	c.1944_1948del	p.Lys649GlufsTer8	Frameshift	LP (PVS1, PM2)	NM_182931.3	608444
*LARP7* ^#^	c.1118_1130del	p.Val373Glufs*11	Frameshift	LP (PVS1, PM2)	NM_016648.4	612026
*LARS2* *	c.1420del	p.Leu474Trpfs*6	Frameshift	P (PVS1, PS2, PM2)	NM_015340.4	604544
*MN1*	c.97del	p.His33ThrfsTer20	Frameshift	LP (PVS1, PM2)	NM_002430.3	156100
*NAA15* *	c.2214del	p.Met738IlefsTer18	Frameshift	P (PVS1, PS2, PM2)	NM_057175.5	608000
*NACC1* *	c.166C>T	p.Arg56Trp	Missense	LP (PS2, PM2, PP3)	NM_052876.3	610672
*NEXMIF*	c.1998del	p.Glu667Lysfs*5	Frameshift	LP (PVS1, PM2)	NM_001008537.2	300524
*NF1*	c.2056A>T	p.Lys686*	Nonsense	LP (PVS1, PM2)	NM_001042492.2	613113
*NFIB* *	c.626A>G	p.Glu209Gly	Missense	LP (PS2, PM2, PP2, PP3)	NM_001190737.3	600728
*NFIX* *	c.442del	p.Ile148Serfs*71	Frameshift	P (PVS1, PS2, PM2)	NM_001271043.2	164005
*LPM1D* *	c.1245dupT	p.Thr416Tyrfs*18	Frameshift	P (PVS1, PS2, PM2)	NM_003620.4	605100
*PUF60* *	c.1334C>T	p.Thr445Ile	Missense	LP (PS2, PM2, PP3)	NM_014281.5	604819
*RFX7* *	c.2236C>T	p.Gln746*	Nonsense	P (PVS1, PS2, PM2)	NM_022841.7	612660
*RPS6KA3* *	c.383C>T	p.Pro128Leu	Missense	LP (PM2, PM6, PP2, PP3)	NM_004586.3	300075
*SATB2*	c.1165C>A	p.Arg389Ser	Missense	LP (PS1, PM2, PM5, PP3)	NM_001172509.2	608148
*SCN1A* *	c.4987G>T	p.Gly1663Cys	Missense	P (PS2, PM1, PM2, PP2, PP3)	NM_006920.6	182389
*SCN1A* *	c.4582-1_4583del	-	Splicing site	P (PVS1, PS2, PM2)	NM_001165963.4	182389
*SCN1A* *	c.4987G>T	p.Gly1663Cys	Missense	P (PS2, PM1, PM2, PP2, PP3)	NM_006920.6	182389
*SCN2A* ^#^	c.641C>A	p.Ser214*	Nonsense	LP (PVS1, PM2)	NM_001040143.2	182390
*SCN8A* *	c.5235C>A	p.Phe1745Leu	Missense	P (PS2, PM1, PM2, PP2, PP3)	NM_001330260	600702
*SETD1A* *	c.5116C>G	p.Leu1706Val	Missense	LP (PS2, PM2)	NM_014712.3	611052
*SHANK3* *	c.352dup	p.Leu118ProfsTer28	Frameshift	P (PVS1, PM6, PM2)	NM_001372044.2	606230
*SIX3* *	c.221del	p.Pro74Argfs*177	Frameshift	P (PVS1, PM6, PM2)	NM_005413.4	603714
*SPEN* *	c.5485_5486insTTTGAAC	p.Gln1829Leufs*2	Frameshift	P (PVS1, PS2, PM2)	NM_015001.4	613484
*STAG2*	c.1018-2_1018-1delinsTT	-	Splicing site	LP (PVS1, PM2)	NM_001042750.2	300826
*STXBP1* *	c.903-1G>C	-	Splicing site	P (PVS1, PS2, PM2)	NM_001032221.3	602926
*SYNGAP1* *	c.1713_1714delinsAC	p.Trp572Arg	Missense	LP (PS2, PM2, PM5)	NM_006772.2	603384
*SYNGAP1* *	c.1216_1218delins	p.Tyr406Asnfs*4	Frameshift	P (PVS1, PS2, PM2)	NM_006772	603384
*TAOK1*	c.1721dupA	p.Ser575Glufs*28	Frameshift	LP (PVS1, PM2)	NM_020791	610266
*TAOK1* ^#^	c.1489_1492del	p.Asp497Lysfs*42	Frameshift	LP (PVS1, PM2)	NM_020791.4	610266
*TBR1* *	c.893dup	p.His298Glnfs*23	Frameshift	P (PVS1, PS2, PM2)	NM_006593.3	604616
*TCF20* *	c.5047_5054del	p.Pro1683Valfs*34	Frameshift	P (PVS1, PS2, PM2)	NM_005650.3	603107
*TCF4*	c.-20-184_72+815del	-	Splicing site	LP (PVS1, PM2)	NM_001083962.2	602272
*TNPO3* ^#^	c.120+2T>G	-	Splicing site	LP (PVS1, PM2)	NM_012470.4	610032
*TRIP12*	c.3206+1G>T	-	Splicing site	LP (PVS1, PM2)	NM_001348323.3	604506
*TUBB* *	c.1145C>T	p.Ser382Leu	Missense	LP (PS2, PM2, PP2, PP3)	NM_178014.4	191130
*TUBB* *	c.1017C>G	p.Ser339Arg	Missense	LP (PS2, PM2, PP2, PP3)	NM_178014.4	191130
*USP7* *	c.502T>C	p.Ser168Pro	Missense	LP (PS2, PM1, PM2)	NM_003470.2	602519
*WAC* *	c.620del	p.Lys207Serfs*124	Frameshift	P (PVS1, PS2, PM2)	NM_016628.5	615049
*ZMIZ1* *	c.1413+4A>G	-	Splicing site	LP (PS2, PM2)	NM_020338.3	607159

## Data Availability

The data that support the findings of this study are available on request from the corresponding author. The data are not publicly available due to privacy and ethical restrictions.

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
