# Peer review of "A Robust and Comprehensive Study of the Molecular and Genetic Basis of Neurodevelopmental Delay in a Sample of 3244 Patients, Evaluated by Exome Analysis in a Latin Population"

_diagnostics, 2025, doi:10.3390/diagnostics15030376_

Round 1
Reviewer 1 Report
Comments and Suggestions for Authors
Explain how your study helped in the clinical management .outline any personalized therapeutic approach in CUL3 /SHANK gene mutations based on your Gene ontology analysis.
How you categorized pathogenic/LP variants in those that are not reported in any databases especially the missense variants and 9% (17/186) 171 of patients with genes with autosomal recessive (AR) inheritance
Were the dominant novel variants tested in the parents?
Any of your patients had more than one gene related to NDD?
To demonstrate the biological significance of the genes associated with the eight phenotypic groups the eight phenotypic groups- which are the 8 phenotype groups
The etiology of such disorders arises from environmental factors such as malnutrition, perinatal infections, drug misuse or pollution, which may contribute to the risk for these disorders through epigenetic dysregulation and mutations; synaptopathies are also a significant cause of NDD in the context of structural as corticogenesis and functional as synaptic disruption, affecting the plasticity, signalling and disrupting cerebral connectivity characterized by an imbalance between excitatory and inhibitory transmissions.
Too long a sentence, simplify for better understanding
The diagnostic yield was significantly different comparing clinical exome with 13.4% and trio, analyzing the parents (diagnostic rates of 21.2%), conditioning the modification of the genetic algorithms in the diagnosis of different NDD, positioning the use of WES as an initial analysis in this type of patients, and going above directed molecular studies such as the triplet expansion study for the FMR1 gene as well as aCGH
Simplify the sentence
More explanation required for Gene ontology analysis. The Figure needs to be more clear. Reading the letters is very difficult. Where is C in fig 2?
Comments on the Quality of English LanguageThe sentences should be simpler rather than complex sentences for a better understanding
Reviewer 2 Report
Comments and Suggestions for Authors
I have a few minor comments, that the authors need to address before the manuscript will be considered for publication.
1. The authors must clarify the statement made in lines 189-192 concerning Table 2. Both the ‘de-novo’ and ‘non-de-novo’ variants are indicated with ‘*’ in Table 2, which readers can’t differentiate.
2. There are discrepancies in decimal points on several occasions. Some of the numbers have an English version (“.”), while some have a Spanish version (“,”). This needs to be rectified throughout the manuscript.
3. In line 107-108, VUS should be abbreviated for ‘Variant of uncertain significance’ instead of ‘of uncertain clinical significance’.
4. In the figure 1, it is mentioned that ‘148 associated genes’, while in line 165 text body, it is mentioned as ‘139 genes associated…’. Please note the right number throughout the manuscript. In the supplementary file, make a column of unique gene names with serial number for readers to identify the gene names.
5. In lines 198-200, please specify the mutations in their respective genes so readers can instantly identify mutations related to a gene.
Reviewer 3 Report
Comments and Suggestions for Authors
In the article “A Robust and Comprehensive Study of the Molecular and Genetic Basis of Neurodevelopmental Delay in a Sample of 3244 Patients, Evaluated by Exome Analysis in a Latin Population”, the authors analyzed a significant cohort of patients with neurodevelopmental delay using exome sequencing. This article describes the results of an interesting and important study.
The list of references is sufficient and relevant.
Overall, the paper is well written, but there are some comments.
Perhaps the title should be changed, since the phrase “Latin population” implies a large heterogeneous group of populations, whereas the study analyzed only patients from Colombia.
In the Materials and Methods section:
1. It is unclear how many patients in each group were analyzed. It is desirable to put it in the text. It is possible to make a separate table in the supplementary with indication for how many patients from which groups single-, duo-, or trio- sequencing was performed.
2. The use of the Human Gene Mutation Database (HGMD) in addition to the ClinVar database is desirable. As the HGMD database is more relevant.
In the Results section:
1. Latin terms such as “de novo” should be italicized.
2. There was a question about the ACMG criteria for genetic variants identified de novo. What ACMG criteria for classifying were used to categorize these variants (PVS1, PS1 etc.)? It would be desirable to indicate them in Table 2 and possibly in a supplementary.
3. It was stated in the methods that the algorithms used in this paper allow for CNV analysis. Why is this data not presented? If no CNV analysis was performed, then this should be removed from the methods.
In the Discussion section:
1. It is desirable to analyze the overlap of the identified genes with known CNVs associated with тeurodevelopmental вelay . To date, many such articles have been published.
Round 2
Reviewer 3 Report
Comments and Suggestions for Authors
The authors have satisfactorily addressed most of my comments and questions. In particular, the authors added ACMG classification criteria for all variants, including in the Supplementary Table.